# ARE FEW-SHOT LEARNING BENCHMARKS TOO SIMPLE ?

## ABSTRACT

We argue that the widely used Omniglot and *mini*ImageNet benchmarks are too *simple* because their *class semantics* do not vary across episodes, which defeats their intended purpose of evaluating few-shot classification methods. The class semantics of Omniglot is invariably "characters" and the class semantics of *mini*ImageNet, "object category". Because the class semantics are so similar, we propose a new method called *Centroid Networks* which can achieve surprisingly high accuracies on Omniglot and *mini*ImageNet without using *any* labels at meta-evaluation time. Our results suggest that those benchmarks are not adapted for *supervised* few-shot classification since the supervision itself is not necessary during meta-evaluation. The Meta-Dataset, a collection of 10 datasets, was recently proposed as a harder few-shot classification benchmark. Using our method, we derive a new metric, the *Class Semantics Consistency Criterion*, and use it to quantify the difficulty of Meta-Dataset. Finally, under some restrictive assumptions, we show that Centroid Networks is faster and more accurate than a state-of-the-art learning-to-cluster method (Hsu et al., 2018).

## 1 INTRODUCTION

*Supervised few-shot classification*, sometimes simply called *few-shot learning*, consists in learning a classifier from a small number of examples. Being able to quickly learn new classes from a small number of labeled examples is desirable from a practical perspective because it removes the need to label large datasets. Typically, supervised few-shot classification is formulated as meta-learning on episodes, where each episode corresponds to two small sets of labeled examples called *support* and *query* sets. The goal is to train a classifier on the support set and to classify the query set with maximum accuracy.

The Omniglot (Lake et al., 2011) and *mini*ImageNet (Vinyals et al., 2016; Ravi & Larochelle, 2017) benchmarks have been heavily used to evaluate and compare supervised few-shot classification methods in the last few years (Vinyals et al., 2016; Ravi & Larochelle, 2017; Snell et al., 2017; Finn et al., 2017; Sung et al., 2018). Despite their popularity and their important role in pioneering the few-shot learning field, we argue that the Omniglot and *mini*ImageNet benchmarks should not be taken as gold standards for evaluating supervised few-shot classification because they rely on *consistent class semantics* across episodes. Specifically, Omniglot classes always correspond to alphabet characters, while *mini*ImageNet classes always correspond to object categories as defined by the WordNet taxonomy (Miller, 1995; Russakovsky et al., 2015). One consequence is that benchmarks with consistent class semantics have similar class semantics between meta-training and meta-evaluation[1]. Therefore, they are too "easy" because they do not test the ability of supervised few-shot classification methods to adapt to new class semantics.

From an applications perspective, being able to adapt to changing class semantics is a desirable feature. For instance, if the application is to organize users' personal photo gallery, different users might want to sort their personal photo gallery according to the different semantics, such as person identity, place or time.

---

[1]We will use the term meta-evaluation to refer to either meta-validation or meta-testing, i.e. evaluation on the meta-learning validation set or test set, respectively.

From a methodological perspective, we argue that supervised few-shot classification becomes an awkward task in the ideal case where the class semantics are perfectly consistent. Indeed, if the end goal of every episode is to classify the query set according to the same class semantics, do we even *need* the support set to define the classes, once the semantics are learned ? Consider the characters below, extracted from the "Mongolian" alphabet of Omniglot. How would you group the characters below?

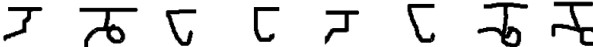

This task is not particularly hard, even if the reader was never shown labeled examples prior to the task, simply because the reader was already familiar with the class semantics of interest (characters), and can generalize them to new classes. This simple observation suggests that when class semantics are consistent, few-shot learning algorithms might not actually need labels during meta-evaluation. To show this, we introduce a new learning-to-cluster[2] method called *Centroid Networks* which achieves surprisingly high accuracies on Omniglot and *mini*ImageNet without using *any* labels at meta-evaluation time.[3] The method is very similar to Prototypical Networks (Snell et al., 2017), but the key difference is that the labels of the support set can be reliably recovered through clustering whenever the cluster semantics are consistent across tasks.

A harder benchmark would involve selecting different cluster semantics across episodes. For example, consider the following set of shapes:

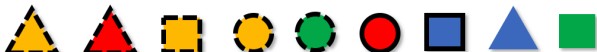

In this case, the task remains ambiguous because clustering semantics (e.g. shape, color, border style) have not been specified. To classify such a set requires either supervision, such as a labeled support set, or to somehow know the class semantics beforehand.

Following that spirit, the Meta-Dataset, a collection of 10 datasets, was recently proposed as a harder and more realistic few-shot classification benchmark (Triantafillou et al., 2019). Among other things such as variable numbers of ways and shots, a key difficulty of the Meta-Dataset is that class semantics vary across episodes, since episodes are generated from a randomly selected dataset. We propose to use Centroid Networks to benchmark how hard this dataset is. In particular, we suggest looking at the *gap* between the performance of Prototypical Networks and Centroid Networks, which we call the *class semantics consistency criterion* (CSCC).

Our contributions:

- We first show that *Centroid Networks*, our proposed approach to perform clustering without labels at meta-evaluation time, can beat a state-of-the-art learning-to-cluster method (Hsu et al., 2018) in the setting of a known number of equally-sized clusters, while being *easier* to train and orders of magnitude *faster* to run.

- We show that it is possible to achieve surprisingly high accuracies on Omniglot and *mini*ImageNet without using *any* labels at meta-evaluation time, using Centroid Networks. This is captured by our proposed metric, *class semantics consistency criterion* (CSCC), which is the first to quantify how easy a few-shot classification benchmark is. This result highlights the need for harder benchmarks which actually test the ability of supervised few-shot classification methods to adapt to new class semantics

- We report CSCC on the recently proposed Meta-Dataset, to assess whether it is indeed a harder benchmark for few-shot classification.

---

[2]When no labels are available, few-shot classification is more naturally formulated as a clustering problem, as detailed in Section 4.

[3]During meta-training, labeled support set is necessary for learning the class semantics. During meta-evaluation, unlabeled support set images are necessary for fair comparison, as detailed in Section 4.

## 2 RELATED WORK

**Supervised clustering.** Supervised clustering is defined in Finley & Joachims (2005) as "learning how to cluster future sets of items [...] given sets of items and complete clusterings over these sets". They use structured SVM to learn a similarity-metric between pairs of items, then run a fixed clustering algorithm which optimizes the sum of similarities of pairs in the same cluster. In follow-up work (Finley & Joachims, 2008), they use K-Means as the clustering algorithm. A main difference with our work is that we learn a nonlinear embedding function, whereas they assume linear embeddings. The work of Awasthi & Zadeh (2010) is also called supervised clustering, although they solve a very different problem. They propose a clustering algorithm which repetitively presents candidate clusterings to a "teacher" and actively requests feedback (supervision).

**Learning to cluster.** Recent deep learning literature has preferred the term "learning to cluster" to "supervised clustering". Although the task is still the same, the main difference is the learning of a similarity metric using deep networks. Because of this aspect, these works are often classified as falling in the "metric learning" literature. Hsu et al. (2017; 2019) propose a Constrained Clustering Network (CCN) for learning to cluster based on two distinct steps: learning a similarity metric to predict if two examples are in the same class, and optimizing a neural network to predict cluster assignments which tend to agree with the similarity metric. CCNs obtained the state-of-the-art results when compared against other supervised clustering algorithms, we will thus use CCN as a strong baseline. In our experiments, Centroid Networks improve over CCN on their benchmarks, while being simpler to train and computationally much cheaper.

**Semi-supervised & constrained clustering.** Semi-supervised clustering consists of clustering data with some supervision in the form of "this pair of points should be/not be in the same cluster". Some methods take the pairwise supervision as hard constraints (Wagstaff et al., 2001), while others (including CCN) learn metrics which tend to satisfy those constraints (Bilenko et al., 2004). See the related work sections in Finley & Joachims (2005); Hsu et al. (2017).

**Supervised few-shot classification.** For the unsupervised few-shot classification task, our method *may* be compared to the supervised few-shot classification literature (Vinyals et al., 2016; Ravi & Larochelle, 2017; Finn et al., 2017). In particular, we have compared with Prototypical Networks (Snell et al., 2017), which was a source of inspiration for Centroid Networks. Our work is also related to follow-up work on Semi-Supervised Prototypical Networks (Ren et al., 2018), in which the support set contains both labeled and unlabeled examples. In this work, we go beyond by requiring no labels to infer centroids at evaluation time.

**Sinkhorn K-Means.** The idea of formulating clustering as minimizing a Wasserstein distance between empirical distributions has been proposed several times in the past (Mi et al., 2018a). Canas & Rosasco (2012) explicit some theoretical links between K-Means and the Wasserstein-2 distance. The most similar work to Sinkhorn K-Means is Regularized Wasserstein-Means (Mi et al., 2018b), but they use another method for solving optimal transport. Specifically using Sinkhorn distances[4] for clustering has even been suggested in Genevay et al. (2018). However, as we could not find an explicit description of the Sinkhorn K-Means anywhere in the literature, we coin the name and explicitly state the algorithm in Section 5.1. To our knowledge, we are the first to use Sinkhorn K-Means in the context of learning to cluster and to scale it up to more complex datasets like *mini*ImageNet. Note that our work should not be confused with Wasserstein K-Means and similar variants, which consist in replacing the squared $L_2$ base-distance in K-Means with a Wasserstein distance.

**Meta-Learning and Unsupervised Learning.** Finally, some recent work has explored combinations of unsupervised learning and meta-learning, to address various other tasks. Metz et al. (2018) propose a method to meta-train an unsupervised representation learning model that produces useful features for some given task. That is, at evaluation time, their method produces features without requiring labels, much like Centroid Networks produce centroids without requiring labels. The difference with their method thus lies in the addressed task: we focus on clustering, while they consider the task of representation/feature learning. Hsu et al. (2018); Khodadadeh et al. (2018) also considers the opposite: meta-learning that requires no labels for meta-training but that delivers methods that require labels to be run at evaluation time. Specifically, they propose unsupervised approaches

---

[4]A regularized version of the Wasserstein distance.

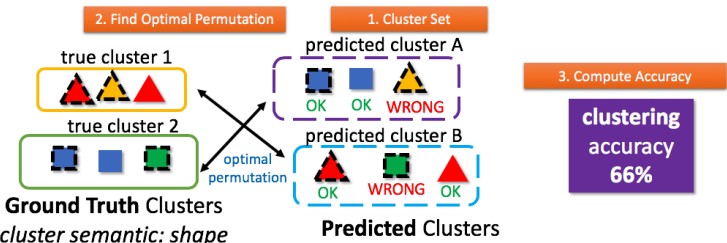

Figure 1: A Few-Shot Clustering Episode.

to generate episodes for supervised few-shot classification, while we use supervised data to learn an unsupervised clustering algorithm.

## 3 TASKS AND EVALUATION METRICS

The main point of this paper is to discuss the class semantics consistency of few-shot classification benchmarks. Recall the visual examples from the introduction, where we asked the reader to cluster similar images together. The more consistent the class semantics are across episodes, the easier it should be to cluster them. Therefore, for the purpose of evaluating semantics consistency, we propose to consider additional categorization tasks for existing few-shot classification benchmarks.

### 3.1 DEFINING OTHER CATEGORIZATION TASKS FROM FEW-SHOT BENCHMARKS

The most common categorization task is supervised few-shot classification, where episodes come with a small training (**support**) set $S = (X_S, Y_S)$ and a small validation (**query**) set $Q = (X_Q, Y_Q)$, where $X_S, X_Q$ denote images or data, and $Y_S, Y_Q$ the associated labels. The task is to predict labels for validation images $X_Q$ and the algorithm has access both to the support set images $X_S$ and labels $Y_S$. Finally, the predicted labels are compared against $Y_Q$, and the accuracy is returned. From now on we call this metric the **supervised accuracy** in order to distinguish it from the clustering and unsupervised accuracy introduced below.

**Few-Shot Clustering Task**    The task is to cluster the query[5] images $X_Q$, without access to the support set $X_S, Y_S$. For evaluation, the predicted clusters are matched with the ground-truth clusters (which can be obtained from $Y_Q$) by searching for the one-to-one ground-truth cluster/predicted cluster mapping (i.e. permutation) which results in the highest accuracy. Finding the optimal permutation can be done efficiently using the Hungarian algorithm as described in Hsu et al. (2017). The resulting accuracy is called the **clustering accuracy**. This is a common metric used in the literature on learning to cluster. See **Figure 1** for an illustration.

Few-shot clustering is the simplest clustering task defined here, and can be seen as an episodic version of the learning to cluster task. However, clustering accuracy cannot be meaningfully compared with supervised accuracy. On one hand, few-shot clustering is harder than supervised few-shot classification because the support set cannot be used. On the other hand, it may be easier because the query set is clustered jointly (vs. independent predictions for supervised few-shot classification). In particular, the 1-shot clustering is trivial because each point already belongs to its own cluster, whereas supervised 1-shot classification is not. Therefore, we propose the unsupervised few-shot classification task which is by construction strictly harder than supervised few-shot classification.

**Unsupervised Few-Shot Classification Task**    The task is to cluster the support set images $X_S$ then to associate each query set image $x_Q$ with one of the predicted clusters. For evaluation, the optimal permutation between predicted clusters and ground- truth clusters (which can be obtained

---

[5]For this description, we cluster the query set to maintain the parallel with supervised few-shot classification evaluation. However, in practice we cluster the support set instead of the query set so that few-shot clustering and unsupervised few-shot classification can share the support set cluster matching step and be more directly comparable.

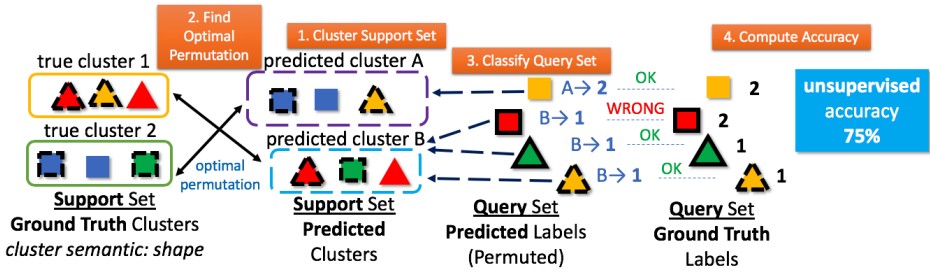

Figure 2: An Unsupervised Few-Shot Classification Episode.

from $Y_S$) is found in order to maximize the corresponding support set accuracy. Then the **unsupervised accuracy** is computed after relabeling the query set predictions and comparing them with $Y_Q$. Unsupervised accuracy can be compared with supervised accuracy because unsupervised is strictly harder than supervised few-shot classification (support set classes $Y_S$ are not available and need to be inferred). See **Figure 2** for an illustration. We will use this metric to define our novel measure for the difficulty of few-shot learning benchmarks.

## 4 BACKGROUND

### 4.1 PROTOTYPICAL NETWORKS

*Prototypical Networks* or *Protonets* (Snell et al., 2017) is one of the simplest and most accurate supervised few-shot classification methods. The only learnable component of Protonets is the embedding function $h_\theta : \mathcal{X} \to \mathcal{Z}$ which maps images to an embedding (feature) space. Given a supervised task $T = (K, M, N, S_{\text{labeled}}, Q)$ to solve, Protonets compute the average embedding (the prototype) of each class $\mu_j = \frac{1}{M} \sum_i h_\theta(x_i^s) * \mathbb{1}\{y_i^s = j\}$ on the support set. Each point from the query set is then classified according to the softmax of its squared distance $p_\theta(y_i^q = j | x_i^q) = \text{softmax}(-||h_\theta(x_i^q) - \mu_j||^2)$. Protonets are trained end-to-end by minimizing the log-loss on the query set.

## 5 CENTROID NETWORKS

In this section, we describe our method and explain how it can be applied to few-shot clustering and unsupervised few-shot classification. *Centroid Networks* (or *CentroidNets*) consist of two modules: a trainable **embedding module** and a fixed **clustering module**. The fact that the only trainable component of Centroid Networks is the embedding function makes implementation and training very simple.

The embedding module is the same as in Prototypical Networks and consists in a neural network $h_\theta : \mathcal{X} \to \mathcal{Z}$ which maps data (images) $x$ to features $z = h_\theta(x)$ in the embedding space. The clustering module takes as input the embedded data $(z_i)_{1 \leq j \leq N}$ and outputs a set of centroids $(c_j)_{1 \leq j \leq K}$ (representatives of each cluster) as well as the (soft) assignment $p_{i,j}$ of each point $z_i$ to each centroid. We use the Sinkhorn K-Means algorithm as our clustering module (Section 5.1).

### 5.1 SINKHORN K-MEANS

We propose *Sinkhorn K-Means* as the clustering module of Centroid Networks. It takes as input a set of $N$ points $x_i \in \mathbb{R}^d$ (typically learned embeddings) and outputs a set of $K$ centroids $c_j \in \mathbb{R}^d$, similarly to K-Means, which can be used to cluster points. Sinkhorn K-Means is based on the Sinkhorn distance (or regularized Wasserstein), described more in depth in Appendix A.1. The differences between Sinkhorn and Regular K-Means, and their formulations as constrained optimization problems are discussed in Appendix B.1. In particular, we expect Sinkhorn K-Means to improve performance on the considered tasks (Section B.2).

**Step 1: Finding Centroids.** We propose an Expectation-Maximization-style procedure to find the centroids which minimize the Sinkhorn distance (Section A.1) between the empirical distributions respectively defined by the data $p(x) = \frac{1}{N}\sum_{i=1}^{N}\delta(x - x_i)$ and the centroids $q(x) = \frac{1}{K}\sum_{j=1}^{K}\delta(x - c_j)$. We alternate descent on assignments and centroids. Minimization in the assignments is a call to the Sinkhorn algorithm (Algorithm 1), instead of the usual greedy argmin for K-Means. Minimization in the centroids amounts to setting them equal to the weighted average of the points assigned to them. For simplicity, Algorithm 2 describes the procedure in the case where clusters are balanced (e.g. Omniglot and *mini*ImageNet). Typically, we initialize centroids around zero and add a tiny bit of Gaussian noise to break symmetries. When clusters are not balanced but the cluster weights are known (e.g. Meta-Dataset), the weights can be passed to the Sinkhorn distance (see Algorithm 1). All details can be found in the code.

---

**Algorithm 1** Sinkhorn$(x, c, \gamma)$ for Wasserstein-2 distance between empirical distributions.

---

**Input:** data $(x_i)_{1 \leq i \leq n} \in \mathbb{R}^{n \times d}$, centroids $(c_j)_{1 \leq j \leq k} \in \mathbb{R}^{k \times d}$, regularization constant $\gamma > 0$.
**Output:** optimal transport plan $(p_{i,j}) \in \mathbb{R}^{n \times k}$.
$K_{i,j} \leftarrow \exp(-||x_i - c_j||_2^2/\gamma)$    // Compute exponentiated pairwise squared L2-distances
$v_j \leftarrow 1$    // Initialize dual variables $(v_j) \in \mathbb{R}^k$
$R_i \leftarrow 1/n, C_j \leftarrow 1/k$    // Set data $(R_i) \in \mathbb{R}^n$ and centroid $(C_j) \in \mathbb{R}^k$ weights to uniform (or desired distribution)
**while** not converged **do**
  $u_i \leftarrow R_i/(\sum_{j=1}^{k} K_{i,j}v_j), \quad 1 \leq i \leq n$    // Enforce row (data) marginals
  $v_j \leftarrow C_j/(\sum_{i=1}^{n} K_{i,j}u_i), \quad 1 \leq j \leq k$    // Enforce column (centroid) marginals
**end while**
$p_{i,j} \leftarrow u_i K_{i,j} v_j, \quad 1 \leq i \leq n, 1 \leq j \leq k$    // Return optimal transport plan.

---

**Algorithm 2** Sinkhorn K-Means$(x, c, \gamma)$

---

**Input:** data $(x_i)_{1 \leq i \leq N}$, initial centroids $(c_j)_{1 \leq j \leq K}$, regularization constant $\gamma > 0$.
**Output:** final centroids $(c_j)_{1 \leq j \leq K}$, optimal assignment $(p_{i,j}) \in \mathbb{R}^{N \times K}$.
**while** not converged **do**
  $(p_{i,j}) \leftarrow \text{Sinkhorn}(x, c, \gamma)$    // Compute OT $p_{i,j}$ between data and centroids
  $c_j \leftarrow k \sum_{i=1}^{n} p_{i,j} x_i, \quad 1 \leq j \leq k$    // Update centroids $c_j$ to minimize cost
**end while**
Return centroids $c$ and assignments $p$.

---

**Step 2: Clustering Points.** Once the centroids are computed, we propose different ways to cluster the data points:

- **Softmax conditionals**. The conditional probability of point $i$ being assigned to centroid $j$ is given by a softmax on their distance: $p_\theta(cluster = j|x = x_i) = \frac{\mathbf{e}^{-||h_\theta(x_i) - c_j||^2/T}}{\sum_{j'=1}^{K} \mathbf{e}^{-||h_\theta(x_i) - c_j||^2/T}}$
  We add an extra temperature parameter $T > 0$. Larger temperatures yield more uniform assignments. This is the way points are classified in Prototypical Networks.

- **Sinkhorn conditionals**. The conditional probability of point $i$ being assigned to centroid $j$ is given by the optimal transport plan $p_{i,j}$ computed previously: $p_\theta(cluster = j|x = x_i) = \frac{p_{i,j}}{\sum_{j'=1}^{K} p_{i,j}}$. Although there is no temperature parameter to tune, the Sinkhorn algorithm has a regularization parameter $\gamma > 0$, which has a similar effect as the temperature, since using both are equivalent to rescaling the distance matrix $||h_\theta(x_i) - c_j||^2$. Using Sinkhorn conditionals favors balanced clusters, but using Softmax conditionals provides no such guarantees.

## 5.2 PREDICTION AND META-EVALUATION

Given a few-shot clustering or unsupervised few-shot classificaiton episode, we embed the raw data $z_i = h_\theta(x_i)$. Then, we cluster the support set in embedding space using Sinkhorn K-Means. Finally,

we associate query set points with predicted clusters by finding their nearest-centroid in embedding space. We compute the clustering and unsupervised accuracies following Section 3.1.

## 5.3 TRAINING WITH A SUPERVISED SURROGATE LOSS (META-TRAINING)

The most intuitive way to train Centroid Networks would be to train them end-to-end, by backprop-agating through Sinkhorn K-Means, which contains two nested loops. Although this is technically possible after defining smoother versions of the clustering/unsupervised accuracies (by replacing the 0-1 loss with a cross-entropy), we did not have much success with this approach. Instead, we opt for the much simpler approach of training with a **supervised surrogate loss**. Since we have access to the ground-truth classes during meta-training, we can simply replace the centroids $c_j$ with the average of each class $\mu_j = \frac{1}{M} \sum_i h_\theta(x_i^s) * \mathbb{1}\{y_i^s = j\}$. Then, we classify the query set points using either Softmax or Sinkhorn conditionals. Finally, we compute the log-loss on the query set and min-imize it using gradient descent.[6] The supervised surrogate loss is very simple, as it removes both the need to find the optimal cluster-class permutation and the the need to backpropagate through Sinkhorn K-means.

**Center Loss.** Additionally to the supervised surrogate loss, we use a center loss penalty (Wen et al., 2016). Center losses have been used in metric-learning methods to penalize the variance of each class in embedding space. See for instance Wen et al. (2016) where it is used in addition to the standard log-loss for learning discriminative face embeddings. Using a center loss makes sense because there is no obvious reason why the surrogate loss (basically a cross-entropy) by itself would make the classes more compact in embedding space. However, compact clusters is an implicit assumption of K-means and Sinkhorn K-means, which makes it essential for having good validation performance. We find experimentally that center loss helps improve clustering and unsupervised accuracies, at the cost of making supervised accuracy slightly worse (we don't use it for training Protonets).

## 5.4 CLASS SEMANTICS CONSISTENCY CRITERION

As a preliminary attempt to quantify how consistent class semantics are across episodes, we define the *Class Semantics Consistency Criterion* as the following ratio:

$$\mathbf{CSCC} := \frac{\text{unsupervised accuracy (Bayes)}}{\text{supervised accuracy (Bayes)}} \approx \widehat{\mathbf{CSCC}} := \frac{\text{unsupervised acc. (Centroid Networks)}}{\text{supervised acc. (Prototypical Networks)}} \quad (1)$$

where we define the supervised and unsupervised Bayes accuracies as the highest possible accura-cies on a given supervised few-shot classification task and its associated unsupervised counterpart. Except for degenerate cases, the CSCC always varies between 0 (classes are totally inconsistent) and 1 (classes are totally consistent). In practice, we approximate the CSCC by replacing the Bayes accuracies with the supervised accuracy of Protonets and the unsupervised accuracy of Centroid Net-works, but with the constraint that their backbone networks have exactly the same architecture. We point out that approximate CSCC is not rigorously defined and can potentially depend significantly on the chosen architecture and hyperparameters. However, we see it as a first step towards quanti-fying the difficulty of few-shot learning benchmarks. The motivations for introducing unsupervised Bayes accuracy and CSCC are discussed more in depth in Sections B.5 and B.4.

## 6 EXPERIMENTS

We first confirm that Centroid Networks is a reasonable approach by comparing it against a state-of-the art few-shot clustering method (Section 6.1). Then, we attempt to use Centroid Networks to evaluate the difficulty (in terms of class semantic variability) of current few-shot learning bench-marks (Sections 6.2 and 6.3). In all cases, we train our method by minimizing the surrogate loss with Softmax conditionals combined with a center loss (both improve accuracies). Our method requires little to no tuning across datasets, and to show this, we run all experiments with the following default hyperparameters: temperature $T = 1$, sinkhorn regularization $\gamma = 1$, center loss of 1, and Sinkhorn

---

[6]When using softmax conditionals, surrogate loss minimization reduces to the standard training of Proto-typical Networks. In fact, the reason for naming our method Centroid Networks is because they can be seen as replacing the Prototypes (class averages) by the Centroids (weighted cluster averages) during prediction.

| **Omniglot** (CCN setting) | $[20, 47]$**-way 20-shot Acc.** |
|---|---|
| *Few-shot Clustering (Clustering Accuracy)* | |
| **K-Means** (raw features) | 21.7%* |
| **CCN (KCL)** (Hsu et al., 2017) | 82.4%* |
| **CCN (MCL)** (Hsu et al., 2019) | 83.3%* |
| **Centroid Networks** (ours, protonet arch.) | **86.8%** $\pm 0.6\%$ |
| **Centroid Networks** (ours, CCN arch.) | **86.6%** $\pm 0.6\%$ |

Table 1: **Top:** Centroid Networks vs. K-Means on raw and Protonet features. **Bottom:** Test clustering accuracies on Omniglot evaluation set, using the Constrained Clustering Network splits (Hsu et al., 2017) (much harder than Ravi splits). Numbers with a star* are those reported in (Hsu et al., 2019). We compared both using the Protonet Conv4 architecture and the architecture in (Hsu et al., 2017) (CCN), which has more filters. The differences between the two architectures are not significant. All our accuracy results are averaged over 1000 test episodes with a fixed model, and are reported with 95% confidence intervals.

| | Omni5-way 5-shot | Omni 20-way 5-shot Acc. | miniINet 5-way 5-shot |
|---|---|---|---|
| | *Few-shot Clustering (Clustering Accuracy)* | | |
| **K-Means** (raw images) | $45.2\% \pm 0.5\%$ | $30.7\% \pm 0.2\%$ | $41.4\% \pm 0.4\%$ |
| **K-Means** (Protonet features) | $83.5\% \pm 0.8\%$ | $76.8\% \pm 0.4\%$ | $48.7\% \pm 0.5\%$ |
| **Centroid Networks** (ours) | **99.6%** $\pm 0.1\%$ | **99.1%** $\pm 0.1\%$ | **64.5%** $\pm 0.7\%$ |

Table 2: **Top:** Few-shot clustering accuracies for Centroid Networks vs. K-Means on raw data and Protonet features.

conditionals for training. The only exceptions are for Omniglot-CCN, where we take center loss weight equal to 0.1, and for the Meta-Dataset, for which we take $\gamma = 0.1$ and a center loss weight of 0.01. Please refer to the Appendix B.6 for an ablation study on the effect of each trick.

## 6.1 SOLVING FEW-SHOT CLUSTERING

We start with experiments designed to validate that Centroid Networks are a competitive approach to learning how to categorize examples without labels (an important assumption behind our proposed CSCC).

**[Table 1]** For this, we consider the specific task of few-shot clustering and compare with Constrained Clustering Networks (Hsu et al., 2017; 2019), a recent state-of-the art learning to cluster method, on the same task as them, which we will denote Omniglot-CCN.[7] Omniglot is resized to $32 \times 32$ and split into 30 alphabets for training (background set) and 20 alphabets for evaluation. The Omniglot-CCN task consists in clustering each alphabet of the evaluation set individually, after training on the background set. This makes it a harder task than standard few-shot classification on Omniglot, because characters from the same alphabet are harder to separate (more fine-grained), and because the number of ways varies from 20 to 47 characters per set. We run Centroid Networks with all default hyperparameters, except a centroid loss of 0.1. The results given in Table 1 show that Centroid Networks outperform all "flavors" of CCN by a margin (86.8% vs. 83.3% highest). Furthermore, Centroid Networks are also simpler and about 100 times faster than CCN, because they require to embed the data only once, instead of iteratively minimizing a KCL/MCL criterion.

However, we wish to point out that Centroid Networks are less flexible than CCNs, as they require specifying the number of clusters and making an assumption on the sizes of the clusters (in our case, equal size). For this reason, CCNs are more appropriate for the general setting where such assumptions cannot be made. That said, Centroid Networks are particularly suited to our CSCC metric for few-shot classification benchmarks, as they are very efficient and otherwise require strictly less information than a supervised few-shot learning method would. Note that extending Centroid Networks to be as flexible as CCNs would be a promising direction for developing new learning-to-cluster methods.

---

[7]We make the choice to apply our method on their task rather than the opposite because their method is much slower and more complicated to run. By solving the same task as them, we can compare directly with the results from their paper.

| | Omni 5-Way 5-shot | Omni 20-way 5-shot | *mini*INet 5-way 5-shot |
|---|---|---|---|
| *With Labels : Supervised Few-shot Classification (Supervised Accuracy)* | | | |
| **Prototypical Networks** | 99.7%* | 98.9%* | 68.7%$\pm$ 0.5% |
| *No Labels : Unsupervised Few-shot Classification (Unsupervised Accuracy)* | | | |
| **Centroid Networks** (ours) | 99.1% $\pm$ 0.1% | 98.1% $\pm$ 0.1% | 55.3%$\pm$ 0.5% |
| Approximate Class Semantics Consistency Criterion | | | |
| **Approximate CSCC** | 99.4% | 99.2% | 80.5%$\pm$ 0.9% |

Table 3: Using Centroid Networks to solve Omniglot and *mini*ImageNet without using meta-testing labels (unsupervised few-shot classification). We compare the unsupervised test accuracy of centroid networks with the supervised test accuracy of Protonets. Centroid Networks can solve Omniglot almost perfectly (CSCC close to 100%), which suggests the class semantics are extremely consistent, while there is a small gap for *mini*ImageNet (CSCC close to 80%), which suggests the class semantics are fairly consistent. Accuracy results are averaged over 1000 test episodes with a fixed model, and are reported with 95% confidence intervals.

[**Table 2**] We also compare Centroid Networks with two baselines on Omniglot and *mini*ImageNet (standard splits, see Appendix A.2). We run K-Means with K-Means++ initialization directly on the raw images and show that it performs very poorly even on Omniglot, which confirms the importance of learning an embedding function. We also run K-Means on pretrained Protonet features, which is a more interesting comparison, since at the highest level, our method could be described as just clustering Protonet embeddings. It turns out that Centroid Networks still outperform K-Means on the embeddings by a substantial margin on both Omniglot (99.6% vs. 83.5% for 5-way) and *mini*ImageNet (64.5% vs. 48.7%), which confirms the importance of combining Sinkhorn K-Means and the center loss trick. [8]

## 6.2 SOLVING OMNIGLOT AND **MINI**IMAGENET WITHOUT USING META-TESTING LABELS

We now come to the main contribution of this work, which is to assess the difficulty of current few-shot learning benchmarks, using CSCC.

[**Table 3**] We report the performance of CentroidNets on unsupervised few-shot classification tasks on Omniglot and *mini*ImageNet. We also report the performance of Prototypical Networks for the standard supervised few-shot classification tasks. This comparison between the models is fair (Section B.3) even if they solve different tasks, because unsupervised few-shot classification is strictly harder than supervised few-shot classification (Section 3.1). Data splits and architectures are the same as in Protonets, and can be found in Section A.2 of the Appendix.[9] For Omniglot, CentroidNets achieves nearly the same accuracy as Prototypical Networks despite using none of the labels of the support set. The CSCCs of 0.99 are nearly equal to the maximum, which supports our hypothesis that Omniglot has nearly perfect class semantics consistency. For *mini*ImageNet CentroidNets can still achieve an unsupervised accuracy of 55.3%, which is of the same order as the supervised accuracy of 68.7%, despite not using any labels from the support set. The CSCC of 0.80 is not as high as Omniglot but still suggests that there is still a fairly high amount of class semantics consistency.

## 6.3 USING CSCC TO EVALUATE THE DIFFICULTY OF META-DATASET

[**Table 4**] We use approximate CSCC to evaluate the difficulty of Meta-Dataset, under the two settings presented in Triantafillou et al. (2019): *meta-train on ILSVRC* and *meta-train on all datasets*. Traffic Sign and MSCOCO are evaluation-only datasets which are excluded from the meta-training. Meta-evaluation is done on all datasets. We use the same Resnet-18 architecture and hyperparam-

---

[8]Interestingly, on Omniglot 20-way 5-shot, the clustering accuracy of Centroid Networks is actually a bit higher than the supervised accuracy of Protonets (99.1% vs. 98.9%) despite using no labels from the support set. Although impressive, this result is not paradoxical and only confirms that clustering accuracies cannot be directly compared with supervised accuracies (Section 3.1).

[9]Note that we exclude the 1-shot setting from our experiments because it is trivial in our case. For unsupervised few-shot classification, the centroids would be equal to the prototypes up to permutation, and Centroid Networks would reduce to Prototypical network (for evaluation).

|  | Train only on ILSVRC | | | Train on All Datasets | | |
|---|---|---|---|---|---|---|
| **Test Source** | **ProtoNets** | **CentroidNets** | **ApproxCSCC** | **ProtoNets** | **CentroidNets** | **ApproxCSCC** |
| ILSVRC | 44.12±1.08 | 26.40±0.88 | 59.83±2.47 | 41.79±1.04 | 23.84±0.82 | 57.06±2.42 |
| Omniglot | 53.40±1.33 | 36.83±1.20 | 68.96±2.84 | 81.93±0.97 | 66.25±1.12 | 80.86±1.67 |
| Aircraft | 45.29±0.91 | 24.15±0.72 | 53.33±1.91 | 69.43±0.89 | 57.50±1.01 | 82.82±1.81 |
| Birds | 63.59±0.94 | 41.08±1.05 | 64.61±1.91 | 64.73±0.97 | 43.56±1.03 | 67.29±1.89 |
| Textures | 61.78±0.77 | 39.63±0.70 | 64.15±1.39 | 66.35±0.73 | 43.50±0.76 | 65.57±1.35 |
| QuickDraw | 49.58±1.06 | 31.04±0.95 | 62.60±2.33 | 67.74±0.94 | 46.96±1.04 | 69.32±1.81 |
| Fungi | 35.27±1.06 | 18.11±0.71 | 51.34±2.53 | 38.94±1.08 | 21.76±0.76 | 55.89±2.49 |
| VGG Flower | 78.09±0.82 | 47.98±0.96 | 61.44±1.39 | 84.45±0.74 | 55.11±0.95 | 65.26±1.26 |
| Traffic Sign | 46.08±1.10 | 22.03±0.66 | 47.82±1.84 | 49.91±1.01 | 22.71±0.66 | 45.50±1.61 |
| MSCOCO | 35.63±1.03 | 18.19±0.69 | 51.05±2.42 | 36.64±1.04 | 17.60±0.77 | 48.04±2.50 |

Table 4: Using Centroid Networks to solve Meta-Dataset without using meta-testing labels (unsupervised few-shot classification) under the two originally proposed settings : *training on ILSVRC*, and *training on all datasets except Traffic Sign and MSCOCO*. We report supervised test accuracy for Prototypical Networks (reproduced from the official implementation), unsupervised test accuracy for Centroid Networks (ours), and approximate CSCCs (their ratio). All numbers are in percentages, all accuracies are averaged over 600 test episodes.

eters for Protonets and CentroidNets, but CentroidNets is trained with an additional of center loss of 0.001 during meta-training. We pretrain on ILSVRC for the *all datasets* setting. The first observation is that supervised/unsupervised accuracies and approximate CSCCs,[10] are higher when training on all datasets instead of training on ILSVRC only, except for ILSVRC (since it is used in both trainings), Traffic Sign and MSCOCO. The fact that CSCC for Traffic Sign and MSCOCO is actually lower when training on more datasets either means that training on ILSVRC alone can sometimes be better for transfer learning, or is a consequence that the sampling scheme is not optimal (Triantafillou et al., 2019). Aircraft (53%) and Omniglot (69%) are the ones that benefit the most from training on all datasets in terms of CSCCs. We compare approximate CSCCs inside each sub-table.[11] High CSCCs in the *all datasets* sub-table suggest that those datasets have very self-consistent class semantics : Omniglot and Aircraft both have very high CSCCs (more than 80%), while ILSVRC (57%) and Fungi (56%) have the lowest ones. It is less clear how to interpret the CSCCs in the *only ILSVRC* sub-table, but High CSCCs might suggest that those datasets have very similar class semantics with ILSVRC. Except Omniglot (69%), most datasets have fairly low CSCCs. It is interesting to note that some datasets higher CSCC than ILSVRC itself. We leave to future work to determine whether it means that ILSVRC is so inconsistent that it is easier to adapt to other datasets, or if it is a shortcoming of our metric.

## 7    CONCLUSION

We proposed Centroid Networks for performing clustering without labels at meta-evaluation time, and with the idea of using it to assess the difficulty of few-shot classification benchmarks. First, we validate our method by beating a state-of-the-art few-shot clustering method (Hsu et al., 2018) in the setting of a known number of equally-sized clusters, with the advantage that our method is *easier* to train and orders of magnitude *faster* to run. Then, we define the CSCC metric from the unsupervised accuracy of Centroid Networks, and use it for quantifying the difficulty of current few-shot learning benchmarks in terms of class semantics consistency. We find that Omniglot has extremely consistent class semantics (CSCC close to 1), and that *mini*ImageNet has fairly high CSCC as well (CSCC close to 0.8), which backs the intuition that its class semantics invariably correspond to object categories. Our results on the Meta-Dataset benchmark show that it has much lower CSCCs than Omniglot in all settings, and lower CSCCs than *mini*ImageNet in the *ILSVRC only* setting, which confirms that

---

[10]Note that Meta-Dataset has the particularity that meta-training and meta-testing consist of different groups of datasets, therefore CSCC does not only correspond to the class semantics consistency across episodes, but also accounts for the class semantics shift between training and testing.

[11]It is unclear whether the values of the Meta-Dataset approximate CSCCs can be directly compared to Omniglot and *mini*ImageNet because its number of ways and shots are variable, and because Meta-Dataset has a discrepancy between meta-training and meta-evaluation distributions which is not the case for those datasets.

Meta-Dataset has harder and more diverse class semantics. As future work, we would like to improve the CSCC by making it more interpretable and less dependent on the backbone architectures.

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

# A    APPENDIX : BACKGROUND AND IMPLEMENTATION DETAILS

## A.1    SINKHORN DISTANCES

The *Wasserstein-2* distance is a distance between two probability masses $p$ and $q$. Given a base distance $d(x, x')$, we define the cost of transporting one unit of mass from $x$ to $x'$ as $d(x, x')^2$. The Wasserstein-2 distance is defined as the cheapest cost for transporting all mass from $p$ to $q$. When the transportation plan is regularized to have large entropy, we obtain Sinkhorn distances, which can be computed very efficiently for discrete distributions (Cuturi, 2013; Cuturi & Doucet, 2014) (entropy-regularization makes the problem strongly convex). Sinkhorn distances are the basis of the Sinkhorn K-Means algorithm, which is the main component of Centroid Networks. In Algorithm 1, we describe the Sinkhorn algorithm in the particular case where we want to transport mass from the weighted data points $(x_i, R_j)$ to the weighted centroids $(c_j, C_j)$, where $R_j$ and $C_j$ are the weights of the data points and centroids, respectively. In practice, we leverage the log-sum-exp trick in the to avoid numerical underflows.

## A.2    DATA SPLITS AND ARCHITECTURE FOR OMNIGLOT AND *mini*IMAGENET EXPERIMENTS

For the embedding network for the Omniglot and *mini*ImageNet, we reuse exactly the same simple convolutional architecture as in Prototypical Networks (Snell et al., 2017), which consists of four stacked blocks (2D convolution with $3 \times 3$ kernel and stride 1, BatchNorm, ReLU, and $2 \times 2$ max-pooling), the output of which is flattened. This results in a 64-dimensional embedding for Omniglot and 1600-dimensional embedding for *mini*ImageNet. For *mini*ImageNet, we pretrain the embedding function using prototypical networks to solve 30-way problems instead of 5, which is the recommended trick in the paper (Snell et al., 2017). For the other settings, we train from scratch.

Omniglot (Lake et al., 2011) consists of a total of 1623 classes of handwritten characters from 50 alphabets, with 20 examples per class. Images are grayscale with size $28 \times 28$. We follow the same protocol as in Prototypical Networks and use the "Vinyals" train/validation/test splits. We consider 5-way 5-shot and 20-way 5-shot settings (15 query points per class).

*mini*ImageNet (Vinyals et al., 2016) consists of 100 classes, each containing 600 color images of size $84 \times 84$. We follow the "Ravi" splits: 64 classes for training, 16 for validation, and 20 for testing. We consider the 5-way 5-shot setting (15 query points per class).

# B    APPENDIX : QUESTIONS AND ANSWERS

## B.1    WHAT IS THE DIFFERENCE BETWEEN SINKHORN K-MEANS VS. REGULAR K-MEANS ?

We present our version of the Sinkhorn K-Means optimization problem, and compare it with regular K-Means. Both of them can be formulated as a joint minimization in the centroids $c_j \in \mathbb{R}^d$ (real vectors) and the assignments $p_{i,j} \geq 0$ (scalars) which specify how much of each point $x_i$ is assigned to centroid $c_j$:

- **K-Means.** Note that compared to the usual convention, we have normalized assignments $p_{i,j}$ so that they sum up to 1.

$$
\begin{aligned}
\text{minimize} \quad & \min_{p,c} \sum_{i=1}^{N} \sum_{j=1}^{K} p_{i,j} \|x_i - c_j\|^2 \\
\text{subject to} \quad & \sum_{j=1}^{K} p_{i,j} = \frac{1}{N}, \qquad i \in 1{:}N \\
& p_{i,j} \in \{0, \tfrac{1}{N}\}, \qquad i \in 1{:}N, \; j \in 1{:}K
\end{aligned}
$$

- **Sinkhorn K-Means.**

$$\text{minimize} \quad \min_{p,c} \sum_i \sum_j p_{i,j} ||x_i - c_j||^2 - \gamma \underbrace{H(p)}_{\text{entropy}}$$

$$\text{subject to} \quad \sum_{j=1}^{K} p_{i,j} = \frac{1}{N}, \qquad i \in 1:N$$

$$\sum_{i=1}^{N} p_{i,j} = \frac{1}{K}, \qquad j \in 1:K$$

$$p_{i,j} \geq 0 \qquad i \in 1:N, \ j \in 1:K$$

where $H(p) = -\sum_{i,j} p_{i,j} \log p_{i,j}$ is the entropy of the assignments, and $\gamma \geq 0$ is a parameter tuning the entropy penalty term.

**Sinkhorn vs. Regular K-Means.** The first difference is that K-Means only allows hard assignments $p_{i,j} \in \{0, \frac{1}{N}\}$, that is, each point $x_i$ is assigned to exactly one cluster $c_j$. On the contrary, the Sinkhorn K-Means formulation allows soft assignments $p_{i,j} \in [0, \frac{1}{N}]$, but with the additional constraint that the clusters have to be balanced, i.e., the same amount of points are soft-assigned to each cluster $\sum_i p_{i,j} = \frac{1}{K}$. The second difference is the penalty term $-\gamma H(p)$ which encourages solutions of high-entropy, i.e., points will tend to be assigned more uniformly over clusters, and clusters more uniformly over points. Adding entropy-regularization allows us to compute $p_{i,j}$ very efficiently using the work of Cuturi (2013). Note that removing the balancing constraint $\sum_i p_{i,j} = \frac{1}{K}$ in the Sinkhorn K-Means objective would yield a regularized K-Means objective with coordinate update steps identical to EM in a mixture of Gaussians (with $p_{i,j}$ updated using softmax conditionals).

### B.2 Why is Sinkhorn K-means expected to improve performance ?

The ablation study in Section B.6 shows that using Sinkhorn K-Means instead of K-Means is the most decisive factor in improving performance. There are mainly two possible explanations :

1. Sinkhorn K-Means is particularly well adapted to the few-shot clustering and unsupervised few-shot classification problems because it strictly enforces the fact that the classes have to follow a given distribution (e.g. balanced), whereas K-Means does not.

2. Sinkhorn K-Means is likely to converge better than K-means due to the entropy-regularization factor of the Sinkhorn distance.

To illustrate the second point, consider the limit case where the regularization factor of Sinkhorn distance goes to infinity ($\gamma \to \infty$). Then, the assignments in Sinkhorn K-Means become uniform (each cluster is assigned equally to all points), and all the centroids converge – in one step – to the average of all the points, reaching global minimum. This is by no means a rigorous proof, but the limit case suggests that Sinkhorn K-Means converges well for large enough $\gamma$. This behavior is to be contrasted with K-means, for which convergence is well known to depend largely on the initialization.

### B.3 What is the effect of using weighted vs. unweighted averages ?

One could argue that comparing CentroidNets with ProtoNets is unfair because using Sinkhorn K-Means leads to centroids which are weighted averages, whereas ProtoNet prototypes are restricted to be unweighted averages. Therefore, we run Centroid Networks on *mini*Imagenet, but under the constraint that centroids to be unweighted averages of the data points. To do so, starting from the soft weights, we reassign each data point only to its closest centroid, and compute the unweighted averages. The comparison between ProtoNets and CentroidNets is now fair in the sense that both prototypes and centroids use unweighted averages.

- Unsupervised accuracy on *mini*Imagenet is $0.5508 \pm 0.0072$ for weighted average and $0.5497 \pm 0.0072$ for unweighted average. The difference is not significant.

- Clustering accuracy on *mini*Imagenet is $0.6421 \pm 0.0069$ for weighted average and $0.6417 \pm 0.0069$ for unweighted average. The difference is also not significant.

This experiment suggests that using weighted averages does not bring an unfair advantage, and therefore does not invalidate our comparison. More generally, instead of trying to tune ProtoNets and CentroidNets as well as possible, we try to make ProtoNets and CentroidNets more comparable by using the same architectures and representation.

### B.4    What does Unsupervised Bayes Accuracy quantify ?

We define the unsupervised Bayes accuracy of an unsupervised few-shot classification task distribution as the highest achievable unsupervised accuracy. Just like the usual Bayes error is limited by label noise, the unsupervised Bayes accuracy is limited by cluster-semantic noise of a task.

For illustration, consider the following unsupervised few-shot classification task distribution :

1. Uniformly sample a random dimension $1 \leq j \leq D$ (hidden to the algorithm)
2. Sample (iid, with probability $1/2$) random binary vectors $(x_i)_{1 \leq i \leq D}$ of dimension $D$ (shown to the algorithm) and split them between support and query set.
3. Assign binary labels $y = x_j$ to each vector $(x_i)$ (hidden to algorithm).
4. The goal is to cluster the support set and associate query set points with the support clusters.

Because the algorithm does not know which dimension $j$ was sampled (i.e. the class semantic), it does not know how to cluster the support set. Therefore, it is just as good to make random predictions on the query set. Therefore the unsupervised Bayes accuracy is $0.5$.

Now, consider the same task distribution, except the dimension index $j$ is always fixed to 1. After meta-training, the algorithm can learn a representation mapping each vector to the value of its first dimension only. The support set can be clustered by grouping all 1s together, and all 0s together. Each query point can then be unambiguously assigned to one of the clusters. The resulting unsupervised Bayes accuracy is $1$.

Both task distributions would become equivalent if the algorithm had access to the class semantics $j$. Therefore, the two unsupervised few-shot tasks differ in difficulty only because of the uncertainty/variability on class semantics, and this is reflected in the difference in unsupervised Bayes accuracy.

### B.5    Is CSCC a good proxy for benchmark difficulty ?

CSCC attempts to quantify the importance of the supervision information, which is not directly related to the difficulty of few-shot learning problem. Indeed, the difficulty of few-shot learning problems can come from many aspects, including but not limited to :

- visual difficulty (how hard is it to train a classifier on all the classes at the same time)
- class semantic consistency (how much do the class semantics vary)

If the goal is to design meaningful benchmarks for supervised few-shot classification methods, it is important to understand which aspects make those benchmarks difficult. For instance, consider the limit case of a supervised few-shot classification task in which the same 5 classes are sampled over and over again. The visual difficulty might be extremely high (e.g. very fine-grained classification), which might lead people to believe that it is a good benchmark (because it is hard and all methods achieve low accuracies). However, because there is no variability at all in the class semantic consistency, such a benchmark does not evaluate at all the capacity of few-shot methods of adapting to new tasks.

Our intent is not to introduce CSCC as a proxy for task difficulty (supervised accuracy of SOTA models might be fine for that purpose). Rather, we introduce the CSCC as an attempt to decouple the different axes of difficulty. Dividing the unsupervised Bayes accuracy by the supervised Bayes accuracy is a rough way of normalizing away the visual difficulty (which affects both supervised and unsupervised accuracies) and focusing on the supervision information only.

| | Clustering | Train Cond. | Eval Cond. | Center Loss | ClusteringAcc | +/- | UnsuperAcc | +/- |
|------|-----------------|-------------|------------|-------------|---------------|-------|------------|-------|
| O1 | K-means++ | Softmax | Softmax | No | 83.80% | 0.80% | 85.20% | 0.90% |
| O2 | K-means++ | Softmax | Sinkhorn | No | 87.32% | 0.80% | 84.00% | 0.80% |
| O3 | K-means++ | Sinkhorn | Sinkhorn | Yes | 89.80% | 0.70% | 86.00% | 0.90% |
| O4 | Sinkhorn K-means | Sinkhorn | Sinkhorn | No | 99.20% | 0.20% | 98.50% | 0.20% |
| O5 | Sinkhorn K-means | Softmax | Softmax | No | 99.40% | 0.16% | 98.90% | 0.20% |
| O6 | Sinkhorn K-means | Softmax | Softmax | Yes | 99.50% | 0.10% | 99.00% | 0.10% |
| O7 | Sinkhorn K-means | Softmax | Sinkhorn | Yes | 99.50% | 0.10% | 99.00% | 0.10% |
| O8 | Sinkhorn K-means | Sinkhorn | Sinkhorn | Yes | 99.60% | 0.10% | 99.10% | 0.10% |

Figure 3: Omniglot 5-way 5-shot Ablation Study

| | Clustering | Train Cond. | Eval Cond. | Center Loss | ClusteringAcc | +/- | UnsuperAcc | +/- |
|------|-----------------|-------------|------------|-------------|---------------|-------|------------|-------|
| M1 | K-means++ | Softmax | Softmax | No | 50.50% | 0.50% | 39.40% | 0.70% |
| M2 | K-means++ | Softmax | Sinkhorn | No | 57.40% | 0.60% | 38.60% | 0.70% |
| M3 | K-means++ | Softmax | Softmax | Yes | 50.60% | 0.30% | 39.50% | 0.40% |
| M4 | K-means++ | Softmax | Sinkhorn | Yes | 56.90% | 0.30% | 38.70% | 0.40% |
| M5 | K-means++ | Sinkhorn | Softmax | Yes | 48.60% | 0.60% | 36.70% | 0.70% |
| M6 | K-means++ | Sinkhorn | Sinkhorn | Yes | 56.60% | 0.60% | 35.80% | 0.70% |
| M7 | Sinkhorn K-means | Sinkhorn | Sinkhorn | No | 62.30% | 0.70% | 52.50% | 0.80% |
| M8 | Sinkhorn K-means | Softmax | Softmax | No | 63.70% | 0.70% | 54.80% | 0.80% |
| M9 | Sinkhorn K-means | Softmax | Softmax | Yes | 64.50% | 0.80% | 55.30% | 0.80% |
| M10 | Sinkhorn K-means | Softmax | Sinkhorn | Yes | 64.50% | 0.80% | 55.40% | 0.80% |
| M11 | Sinkhorn K-means | Sinkhorn | Sinkhorn | Yes | 63.10% | 0.70% | 53.20% | 0.80% |

Figure 4: *mini*ImageNet 5-way 5-shot Ablation Study

## B.6 ABLATION STUDY

[**Figures 3,4**] We conduct an ablation study on Omniglot (5-way 5-shot) and *miniImageNet* (5-way 5-shot) to determine the effect and importance of the various proposed tricks and components:

- **K-Means vs. Sinkhorn K-Means.** From comparing O3 to O4, O1 to O5, M6 to M7, M1 to M8, it appears that using Sinkhorn K-Means instead of K-Means++ is the most beneficial and important factor.

- **Center Loss**. From comparing O2 to O3, O5 to O6, O4 to O8, M7 to M11, M8 to M9, center loss seems to be beneficial (although the significance is at the limit of the confidence intervals). It is the second most influential factor.

- **Softmax vs. Sinkhorn conditionals** (at meta-training and meta-evaluation time). For training, it is not clear whether using Sinkhorn or Softmax conditionals is beneficial or not. For evaluation, from comparing M1 to M2, M3 to M4, M5 to M6, it seems that Sinkhorn conditionals are better if the metric is clustering accuracy, while Softmax conditionals might be better if the metric is unsupervised accuracy, although the effect seems to be negligible (see how the color patterns are inverted).

