# OpenReview forum: "Are Few-shot Learning Benchmarks Too Simple ?"
_ICLR.cc/2020/Conference — Reject_

### Official Review · AnonReviewer2 · 2019-10-22
**Official Blind Review #2**

**Rating:** 6

**Review:**

This paper introduces a new method for learning to cluster without labels at meta-evaluation time and show that this method does as well as supervised methods on benchmarks with consistent class semantics. The authors propose a new metric for measuring the simplicity of a few-shot learning benchmark and demonstrate that it is possible to achieve high performance on Omniglot and miniImageNet with their unsupervised method, resulting in a high value of this criterion, whereas the Meta-Dataset is much more difficult.

The paper is well written and generally very clear. I appreciate that the authors have highlighted the limitations of both their clustering method (that it requires more assumptions than CCNs) and their benchmark. The centroid method itself seems to draw heavily on pre-existing work, but uses a new similarity metric that improves performance beyond the current state-of-the-art on few-shot clustering tasks.

The authors acknowledge that the approximate CSCC metric they define is not consistent across architectures and hyperparameters. It is also a fairly simple metric, but nonetheless represents a novel contribution.

Overall I feel that the paper introduces a well-defined problem and makes a step toward quantifying and resolving it. The experiments do a thorough job supporting the arguments of the authors.


I have only minor issues that could help with the clarity of this paper:

I wasn’t sure what was meant by “relabeling” the query set predictions in the text below Figure 2.

I would have appreciated some discussion as to why Sinkhorn distance might be expected to improve performance .

**Experience Assessment:**

I do not know much about this area.

**Review Assessment: Checking Correctness Of Derivations And Theory:**

N/A

**Review Assessment: Checking Correctness Of Experiments:**

I assessed the sensibility of the experiments.

**Review Assessment: Thoroughness In Paper Reading:**

I read the paper at least twice and used my best judgement in assessing the paper.

---

> ### Author Response · Authors · 2019-11-13
> **Answer to Reviewer 2**
>
> We thank the reviewer for their positive review and constructive feedback. We have added explanations relating to the advantages of Sinkhorn K-Means to the Appendix.
>
>
> * Relabeling the query set
>
> Any clustering is permutation invariant. Therefore, there are many equally correct ways to label the support set and the query set, which is why we find the optimal permutation which matches the ground truth cluster indices with the predicted cluster indices.
>
> Specifically for Figure 2:
> - cluster indices are predicted for the query set shapes during step 3.
> {yellow square} gets assigned to cluster A, {red square, green triangle, yellow triangle} get assigned to cluster B.
> - the optimal permutation shows that cluster A matches Class 2, and cluster B matches Class 1
> - therefore, {yellow square} gets relabeled to Class 2, {red square, green triangle, yellow triangle} get relabeled to class 1
> - the unsupervised accuracy can be computed by comparing the predicted and ground truth classes 1, 2.
>
>
> * Why is Sinkhorn K-means expected to improve performance ?
>
> There are mainly two reasons why Sinkhorn K-Means improves performance compared to K-Means :
> - Sinkhorn K-Means is particularly well adapted to the few-shot clustering and unsupervised few-shot classification problems because it strictly enforces the fact that the classes have to follow a given distribution (e.g. balanced), whereas K-Means does not.
> - Sinkhorn K-Means is likely to converge better than K-means due to the regularization factor of the Sinkhorn distance.
>
> To illustrate the second point, consider the limit case where the regularization factor of Sinkhorn distance goes to infinity. Then, the assignments in Sinkhorn K-Means become uniform (each cluster is assigned equally to all points), and all the centroids converge to the average of all the points, which in this case is a global minimum. This is by no means a proof, but this example suggests that for large enough regularization, Sinkhorn K-Means will converge better.

---

### Official Review · AnonReviewer1 · 2019-10-22
**Official Blind Review #1**

**Rating:** 3

**Review:**

The paper is concerned with few-shot classification, both its benchmarks and method used to tackle it. The scope of the few-shot classification problem can be set relatively widely, depending on what data is available at what stage. In general few-shot classification is an important ability of intelligent systems and arguably an area in which biological systems outperform current AI systems the most.

The paper makes a number of contributions. (1) It suggests an approach to do a specific type of clustering and compares it favorably to the existing literature. In a specific sense the approach does not use supervised labels (“without labels at meta-evaluation time”). (2) It applies that approach to currently existing datasets and achieves “surprisingly high accuracies” in that setting, with the implication that this shows a weakness in these datasets when used for benchmarking (“too easy”). (3) It further suggests a metric, dubbed “class semantics consistency criterion”, that aims to quantify this shortcoming of current benchmarks on these datasets. (4) It assesses a specific meta-dataset using that metric, confirming it is harder in this sense, at least in specific settings.

My assessment of the paper is mildly negative; however this is an assessment with low confidence given that I am no expert on few-shot classification or related areas.

While the authors first example (the “Mongolian” alphabet of the Omniglot dataset and geometric shapes falling into different categories) illustrates the problem space well and is indeed quite intuitive, the same cannot be said about either the specific setting they consider nor the metric they propose. It’s not immediately clear that the other approaches from the literature they compare their method to were conceived for the setting considered here, or indeed optimized for it. The authors do show good accuracy on clustering Omniglot characters without using labels and thus indeed demonstrate a high amount of class semantics consistency for that dataset. The results on miniImageNet are less clear-cut, and the results of the evaluation of the meta-dataset appear to depend on the specific setting considered. This makes it unclear to what extent the proposed metric is general and predictive. To their credit, the authors state that in future work they are looking to make their metric “more interpretable and less dependent on the backbone architectures”.

I believe the paper might benefit from being given additional attention. A streamlined and more accessible version might well be an important contribution in the future.

**Experience Assessment:**

I do not know much about this area.

**Review Assessment: Checking Correctness Of Derivations And Theory:**

I did not assess the derivations or theory.

**Review Assessment: Checking Correctness Of Experiments:**

I assessed the sensibility of the experiments.

**Review Assessment: Thoroughness In Paper Reading:**

I read the paper at least twice and used my best judgement in assessing the paper.

---

> ### Author Response · Authors · 2019-11-13
> **Answer to Reviewer 1**
>
> We thank the reviewer for taking the time to review our paper. We address the reviewer’s concerns below.
>
>
> * It’s not immediately clear that the other approaches from the literature they compare their method to were conceived for the setting considered here, or indeed optimized for it.
>
> Supervised Few-Shot Classification literature:
> - Our main contribution is to compare CentroidNets on unsupervised few-shot classification vs. ProtoNets on supervised few-shot classification.
> - ProtoNets were conceived and optimized precisely for the supervised few-shot classification problem (often just called few-shot learning).
> - Our comparison is fair because unsupervised few-shot classification is strictly harder than supervised few-shot classification.
>
> Few-shot clustering literature:
> - Our main contribution doesn't lie in our comparison with CCN (Hsu et al. 2017).
> - The comparison is here mostly to confirm that our approach is reasonable (to have a point of comparison).
> - We are honest and open about the fact that our method is less flexible than CCN.
> - See page 8 “However, we wish to point out that Centroid Networks are less flexible than CCNs, as they require specifying the number of clusters and making an assumption on the sizes of the clusters[...]”
>
>
> * The results on miniImageNet are less clear-cut
>
> Indeed, the gap between supervised and unsupervised accuracies is bigger on miniImageNet.
> - This can be due to the higher visual difficulty of miniImageNet.
> - This can also be due to the lower class semantic consistency.
>
> We do point out that the unsupervised accuracy of our method (55.3%) is still impressive, considering that it is almost equal to the performance of earlier supervised few-shot classification methods with the same architecture (56.2% for MatchingNets without fine-tuning).
>
>
> * The results of the evaluation of the meta-dataset appear to depend on the specific setting considered
>
> Just like the usual supervised accuracy in few-shot learning, the unsupervised accuracy and CSCC are dependent on the task distribution considered. Therefore, it is perfectly normal that the numbers are different for Meta-Dataset in the [Train on ILSVRC] vs. [Train on All datasets] setting, because they define different task distributions (and are therefore different benchmarks).
>
>
> * This makes it unclear to what extent the proposed metric is general and predictive.
>
> In order to better address the reviewer’s concern, we ask the reviewer to clarify what they mean exactly by “general” and “predictive”. Maybe with a concrete example illustrating what these properties are ?

---

### Official Review · AnonReviewer3 · 2019-10-23
**Official Blind Review #3**

**Rating:** 3

**Review:**

<Paper summary>
The authors argue that the popular benchmark datasets, Omniglot and miniImageNet, are too simple to evaluate supervised few-shot classification methods due to their insufficient variety of class semantics. To validate this, the authors proposed clustering-based meta-learning method, called Centroid Network. Although it does not utilize supervision information during meta-evaluation, it can achieve high accuracies on Omniglot and miniImageNet. The authors also proposed a new metric to quantify the difficulty of meta-learning for few-shot classification.

<Review summary>
Although the main claim of this paper seems correct, it is not sufficiently supported by theory or experiments. My score is actually on the border line, but I currently vote for ``weak reject, because some points in the paper are ambiguous yet. Given clarifications in an author response, I would be willing to increase the score.

<Details>
* Strength
 + The paper is well-organized. Especially, the examples shown in the introduction greatly help understanding of what the authors argue in this paper.
 + A novel study on quantifying the difficulty of meta-learning.
 + The proposed CentroidNet performs well in the experiments.

* Weakness and concerns
 - Does CentroidNet really work without labels during ``meta-validation"? As far as I understand, ground truth clusters of the support set defined by the labels are required to compute the accuracies. Therefore, the labels seem to be required to validate the performance of the model. I think it should be ``meta-test."
 - The authors state ``The most intuitive way to train ..., we did not have much success with this approach" in 5.3, but it is counter-intuitive. If the class semantics are similar among episodes, ``the most intuitive way" should work, because it can learn the common semantics via meta-training. Further discussion about why it does not work is required.
 - The high performance of CentroidNet does not support the claim on the insufficient variety of the class semantics. According to ablation study, adopting Sinkhorn K-means is the most important factor to improve the performance. It means that adopting weighted average like in [R1] can also improve the performance of ProtoNet, which results in substantial difference in the performance between ProtoNet and CentroidNet that can deny the claim.
 - The definition of CSCC is not convincing. First, I could not get the meaning of ``unsupervised Bayes accuracy" (supervised Bayes accuracy means 1 - Bayes error rate, right?). Second, CSCC seems to mainly quantify the importance of the supervision information during meta-learning, which is not directly related to the difficulty of few-shot learning problem. Intuitively, difficult few-shot learning problems should lead to lower supervised Bayes accuracy, which does not necessarily decrease CSCC. Third, what we can induce via comparing CSCC is not clarified in theory. The discussion in 6.3 is too subjective and specific for the case of training with ILSVRC/all datasets.
 - This paper lacks citing some closely related works [R1, R2].

[R1] ``Infinite Mixture Prototypes for Few-Shot Learning," ICML2019
[R2] ``A Closer Look at Few-shot Classification," ICLR2019

* Minor concerns that do not have an impact on the score
 - Another arXiv paper related to this work: ``Rapid Learning or Feature Reuse? Towards Understanding the Effectiveness of MAML"



**Experience Assessment:**

I have read many papers in this area.

**Review Assessment: Checking Correctness Of Derivations And Theory:**

I assessed the sensibility of the derivations and theory.

**Review Assessment: Checking Correctness Of Experiments:**

I assessed the sensibility of the experiments.

**Review Assessment: Thoroughness In Paper Reading:**

I read the paper thoroughly.

---

> ### Author Response · Authors · 2019-11-13
> **Answer to Reviewer 3, Part 1/2**
>
> We thank the reviewer for their thorough review and raising several valid points. We have done our best to answer them and we will improve the main paper accordingly (we have added some points to the appendix already). We hope that the reviewer will reconsider their score if we have addressed their concerns.
>
>
> * "CentroidNets uses a variety of tricks to improve performance, therefore it is unfair to not also consider tricks (such as R1) to improve Protonet performance"
> [Rephrased from Reviewer: “The high performance of CentroidNet does not support the claim on the insufficient variety of the class semantics [...] ]
>
> Indeed, Sinkhorn K-Means is a key component in the performance of CentroidNets. However, it is not obvious that using Sinkhorn K-Means would be an unfair advantage compared to Prototypical Networks, for two reasons :
> - In CentroidNets, we use Sinkhorn k-Means to attempt to recover the hidden class labels, i.e. to infer the ground-truth labels. In contrast, ProtoNets has direct access to the ground-truth labels (which incidentally turn out to be hard assignments and lead to unweighted averages).
> - In CentroidNets, we run Sinkhorn k-Means on representations which were learned with the ProtoNet loss, i.e., they were by construction designed to be averaged without weights.
>
> However, in order to best address the reviewer’s concern, we go further and run new experiments on miniImageNet. This time we constrain the centroids to be unweighted averages of the data points. To do so, starting from the soft weights, we reassign each point only to its closest centroid, and compute the unweighted averages. The comparison between ProtoNets and CentroidNets is now fair in the sense that both prototypes and centroids use unweighted averages.
> - Unsupervised Accuracy on miniImageNet is 0.5508 +/- 0.0072 for weighted average and 0.5497 +/- 0.0072 for unweighted average. The difference is not significant.
> - Clustering Accuracy on miniImageNet is 0.6421 +/- 0.0069 for weighted average and 0.6417 +/- 0.0069 for unweighted average. The difference is also not significant.
> We’ll be happy to add these results to the paper, if the review thinks them valuable.
>
> Therefore, the new experiment suggests that using weighted averages does not bring an unfair advantage, and therefore does not invalidate our comparison. More generally, instead of trying to tune ProtoNets and CentroidNets as well as possible, we try to use comparable models for ProtoNets and CentroidNets (same architecture, nearly same representation).
>
>
> * What is unsupervised Bayes accuracy ?
>
> We define the unsupervised Bayes accuracy of an unsupervised few-shot classification task distribution as the highest achievable unsupervised accuracy. Just like the usual Bayes error is limited by label noise, the unsupervised Bayes accuracy is limited by cluster-semantic noise of a task.
>
> For illustration, consider the following unsupervised few-shot classification task distribution :
> - Uniformly sample a random dimension 1<= j <= D (hidden to the algorithm)
> - Sample (iid, probability=½) random binary vectors (x_i) of dimension D (shown to the algorithm) and split them between support and query set.
> - Assign binary labels y = x_j to each vector (x_i) (hidden to algorithm).
> - The goal is to cluster the support set and associate query set points with the support clusters.
>
> Because the algorithm does not know which dimension j was sampled (i.e. the class semantic), it does not know how to cluster the support set. Therefore, it is just as good to make random predictions on the query set. Therefore the unsupervised Bayes accuracy is 0.5
>
> Now, consider the same task distribution, except the dimension index j is always fixed to 1. After meta-training, the algorithm can learn a representation mapping each vector to the value of its first dimension only. The support set can be clustered by grouping all 1s together, and all 0s together. Each query point can then be unambiguously assigned to one of the clusters. The resulting unsupervised Bayes accuracy is 1.
>
> Both task distributions would become equivalent if the algorithm had access to the class semantics j. Therefore, the two unsupervised few-shot tasks differ in difficulty only because of the uncertainty/variability on class semantics, and this is reflected in the difference in unsupervised Bayes accuracy.
>
> If this example is deemed helpful, we’ll be happy to add it to the paper.

---

> > ### Author Response · Authors · 2019-11-13
> > **Answer to Reviewer 3, Part 2/2**
> >
> > * "CSCC attempts to quantify the importance of the supervision information, which is not directly related to the difficulty of few-shot learning problem"
> >
> > Indeed, the reviewer is absolutely right that the difficulty of few-shot learning problems can come from many aspects, including but not limited to :
> > - visual difficulty (~how hard is it to train a classifier on all the classes at the same time)
> > - class semantic consistency (~how much do the class semantics vary)
> >
> > If the goal is to design meaningful benchmarks for supervised few-shot classification methods, it is important to understand which aspects make those benchmarks difficult. For instance, consider the limit case of a supervised few-shot classification task in which the same 5 classes are sampled over and over again. The visual difficulty might be extremely high (e.g. very fine-grained classification), which might lead people to believe that it is a good benchmark (because it is hard and all methods achieve low accuracies). However, because there is no variability at all in the class semantic consistency, such a benchmark does not evaluate at all the capacity of few-shot methods of adapting to new tasks.
> >
> > Our intent is not to introduce CSCC as a proxy for task difficulty (supervised accuracy of SOTA models might be fine for that purpose). Rather, we introduce the CSCC as an attempt to decouple the different axes of difficulty. Dividing the unsupervised Bayes accuracy by the supervised Bayes accuracy is a rough way of normalizing away the visual difficulty (which affects both supervised and unsupervised accuracies) and focusing on the supervision information only.
> >
> >
> > * Why does end-to-end training not work ?
> > [Rephrased from Reviewer: ``the most intuitive way" should work, because it can learn the common semantics via meta-training]
> >
> > It is true that for a fixed image distribution, the general difficulty of the clustering task gets easier as class semantics become more similar. But here, we mean specifically that the issue with meta-training with an end-to-end loss is an *optimization* issue.
> >
> > Indeed, to solve few-shot clustering end-to-end, the first step would be to define a differentiable clustering loss that we can optimize with gradient descent:
> >
> > Loss(\theta) = ClusteringLoss( ClusteringAlgorithm( h_\theta( support_images ) ) )
> >
> > The first challenge is to ensure that ClusteringLoss and SinkhornKMeans are differentiable with respect to their inputs. This is not trivial but doable (we can give more details if needed).
> >
> > However, making ClusteringAlgorithm and ClusteringLoss differentiable does not guarantee that they are smooth enough, and in fact they might be highly nonlinear. For instance, a small perturbation on \theta could lead to completely different final centroids. Because all gradient descent proofs require some sort of regularity on functions, lack of smoothness might be the reason why end-to-end training does not work with CentroidNets.
> >
> > More generally, the idea of replacing or combining the end-to-end loss with an auxiliary loss is not new and has been proposed and successfully implemented many times in the literature. See for instance :
> > - "Hierarchical Graph Representation Learning with Differentiable Pooling" (Section 3.3 of Ying et al 2018). They recognize that "it can be difficult to train the pooling GNN using only [the end-to-end loss]" and propose to use an "an auxiliary link prediction objective".
> > - "Learning Longer-term Dependencies in RNNs with Auxiliary Losses" (Trinh et al. 2018) which proposes auxiliary losses to alleviate the usual problems of BPTT (e.g. gradient vanishing) when only minimizing the end-to-end sequence-to-sequence loss.
> > - More generally, "Limits of End-to-End Learning" (Glasmachers 2017) state that "end-to-end learning can be very inefficient for training neural network models composed of multiple non-trivial modules", and characterize pros and cons of gradient-based end-to-end learning.
> >
> >
> > * Does CentroidNet really work without labels during ``meta-validation ?
> >
> > This seems to be a terminology issue. The existing terminology is ambiguous (for instance “testing/evaluation/validation” could either refer to [make predictions on new data] or [make predictions on new data + compute evaluation metrics]).
> > Our point is that after the meta-training phase, our method does not need any labels to cluster new support/query sets, as opposed to Prototypical Networks which does require the labels of the support set.
> > Any evaluation generally requires some notion of ground truth. The reviewer is correct that the labels are required to compute accuracies. This is standard in the learning to cluster literature (see for instance Hsu et al 2018).

---

> > > ### Comment · AnonReviewer3 · 2019-11-14
> > > **Thank you for your response**
> > >
> > > Thank the authors for the response and additional experiments.
> > >
> > > * "CentroidNets uses a variety of tricks to improve performance, therefore it is unfair to not also consider tricks (such as R1) to improve Protonet performance"
> > > * What is unsupervised Bayes accuracy ?
> > > * Does CentroidNet really work without labels during meta-validation ?
> > > * Why does end-to-end training not work ?
> > >
> > > The responses for the above questions almost make sense to me. Minor concerns are:
> > > - It would be better to show a mathematical definition of unsupervised Bayes accuracy.
> > > - It would be better to show which part is meta-training/meta-validation/meta-test in Fig. 1 and 2.
> > >
> > >
> > > * "CSCC attempts to quantify the importance of the supervision information, which is not directly related to the difficulty of few-shot learning problem"
> > >
> > > My most major concern is related to this question. The authors' main claim in this paper seems to be "since CentroidNet works better than ProtoNet, this dataset has too high class-semantic consistency." However, this is not sufficiently convincing to me, because there are much more factors that can affect the performance of the meta-learning methods as the authors mentioned. To validate this claim, there is most straightforward way: using the same dataset and meta-learning method, we can only change how to generate episodes. For example, in the harder dataset shown in section 1, we can compare "using all semantics to generate episodes" and "using only single semantic to generate episodes" as similar with the example shown in the authors' response. The comparison between CentroidNet and ProtoNet does not directly support the claim.

---

> > > > ### Author Response · Authors · 2019-11-14
> > > > **Answer to “there are much more factors that can affect the performance of the meta-learning methods [...]”**
> > > >
> > > >
> > > > Reviewer: "there are much more factors that can affect the performance of the meta-learning methods [...]”
> > > >
> > > > We agree that the approximate CSCC is not invariant to the backbone architecture :
> > > > -  Nevertheless, we think it is reasonable to say that the unsupervised accuracies for CentroidNets are surprisingly high for Omniglot and quite decent for miniImageNet.
> > > >
> > > > Moreover, in order to limit the factors that can affect the performance of the models we have taken the following precautions:
> > > > - We have used ProtoNets, one of the top methods for the Conv-4 architecture on Omniglot and miniImageNet. This ensures that the estimated unsupervised Bayes accuracy is not too low.
> > > > - We have run CentroidNets on exactly the representation learned by ProtoNets, with the sole difference that we use center loss during training (we do not use it for ProtoNet because it degrades performance very slightly). This ensures that CentroidNets has no unfair advantage over ProtoNets.
> > > >
> > > > Finally, we emphasize the fact that the CSCC metric itself is well defined. The issue lies only in its practical approximation. There are other examples of well defined quantities in statistics for which approximators are highly dependent on the backbone architecture:
> > > > - for instance, mutual information is approximated by neural networks in very high dimensions “MINE: Mutual Information Neural Estimation” (Belghazi 2018). However, the approximation is completely dependent on the neural network considered.
> > > > - the Wasserstein distance is approximated by neural networks in high dimensions “Wasserstein GAN” (Arjovsky 2017), but the approximation also depends on the architecture of the neural network.

---

> > > > > ### Author Response · Authors · 2019-11-14
> > > > > **Mathematical Definitions of Episodic Bayes Accuracies**
> > > > >
> > > > > The reviewer asked for a mathematical definition of unsupervised Bayes accuracy.
> > > > >
> > > > > Please note that the unsupervised and supervised Bayes accuracies we use for CSCC are defined in the episodic setting, which makes them different from the usual Bayes accuracy defined in the batch setting (usual training).
> > > > >
> > > > > -> Batch Supervised Bayes Accuracy (1 - Bayes error)
> > > > >
> > > > > $$BatchSBA = E_{x,y\sim p} [ \mathbf 1\lbrace y = \arg\max_{y'} p(y'|x) \rbrace ]$$
> > > > >
> > > > > this can be rewritten as a sup over all functions $h:X\to Y$
> > > > >
> > > > > $$BatchSBA = \sup_{h:X\to Y} \mathbf E_{x,y\sim p}  \mathbf 1\lbrace y = h(x) \rbrace $$
> > > > >
> > > > > Just like the Bayes accuracy is the best possible accuracy on a classification task, the episodic unsupervised (resp. supervised) Bayes accuracies are the best possible unsupervised (resp. unsupervised) supervised accuracies on a unsupervised (resp. supervised) few-shot classification task. They are all well defined concepts, but we do provide (cumbersome) mathematical definitions below.
> > > > >
> > > > > -> Episodic Supervised Bayes Accuracy
> > > > >
> > > > > Denote $T$ the task, $S\in\mathcal S$ a labeled support set, $(x,y)$ a labeled query point. episodic supervised Bayes accuracy can be written as a sup over all functions $h:\mathcal S\times X\to Y$.
> > > > >
> > > > > $$EpisodicSBA = \sup_{h:\mathcal S\times X\to Y} \mathbf E_{T\sim p(T)} \mathbf E_{S,x,y\sim p(S,x,y|T)} \mathbf 1\lbrace y = h(x, S) \rbrace $$
> > > > >
> > > > > -> Episodic unsupervised Bayes accuracy
> > > > >
> > > > > Episodic unsupervised Bayes accuracy can also be written as a sup over all functions $h:\mathcal S_x\times X\to \mathcal S_y\times Y$ which take as input a unlabeled support set $S_x$, a query point $x$, and predicts a cluster index $y$ and cluster indices for the support set $\widehat S_y$. We define an intermediate operator $F_\sigma: \mathcal S_y\times \mathcal S_y \to \Sigma$ which returns the optimal permutation between predicted clusters and ground truth classes (it's the argmax of the resulting accuracy).
> > > > >
> > > > > Then $EpisodicUBA$ is the solution of the following optimization problem :
> > > > >
> > > > > Maximize w.r.t. $h:\mathcal S_x\times X\to \mathcal S_y\times Y$:
> > > > >
> > > > > $$\mathbf E_{T\sim p(T)} \mathbf E_{S,x,y\sim p(S_x,S_y,x,y|T)} \mathbf 1\lbrace y = \sigma(\widehat y) \rbrace $$
> > > > >
> > > > > Subject to $ (\widehat S_y,\widehat y) = h(S_x, x)$ and $\sigma = F_\sigma(S_x, \widehat S_y)$

---

### Decision · Program_Chairs · 2019-12-19

**Decision:**

Reject

**Comment:**

The paper is interested in assessing the difficulty of popular few-shot classification benchmarks (Omniglot and miniImageNet). A clustering-based meta-learning method is proposed (called Centroid Network), on which a metric is built (gap between the performance of Prototypical Networks and Centroid Networks). As noted by several reviewers, the proposed metric (critical for the paper) is however not motivated enough, nor convincing enough - after discussion, the logic in the metric reasoning seems to remain flawed.